# Universal and tunable liquid–liquid separation by nanoparticle-embedded gating membranes based on a self-defined interfacial parameter

Xiangyu Li [1], Jingjing Liu[1], Ruixiang Qu[1], Weifeng Zhang[2], Yanan Liu[1], Huajun Zhai[1], Yen Wei[1], Hanshi Hu[1] & Lin Feng [1]✉

Superwetting porous membranes with tunable liquid repellency are highly desirable in broad domains including scientific research, chemical industry, and environmental protection. Such membranes should allow for controllable droplet bouncing or spreading, which is difficult to achieve for low surface energy organic liquids (OLs). Here we develop an interfacial physical parameter to regulate the OL wettability of nanoparticle-embedded membranes by structuring synergistic layers with reconfigurable surface energy components. Under the tunable solid-liquid interaction in the aggregation-induced process, the membranes demonstrate positive/negative liquid gating regularity for polar protic liquids, polar aprotic liquids, and nonpolar liquids. Such a membrane can be employed as self-adaptive gating for various immiscible liquid mixtures with superior separation efficiency and permeation flux, even afford successive achievement of high-performance in situ extraction-back extraction coupling. This study should provide distinctive insights into intrinsic wetting behaviors and have pioneered a rational strategy to design high-performance separation materials for diverse applications.

[1] Department of Chemistry, Tsinghua University, 100084 Beijing, P.R. China. [2] Hangzhou Innovation Research Institute of Beihang University, 310051 Hangzhou, P.R. China. ✉email: fl@mail.tsinghua.edu.cn

Artificial surfaces with special wetting behavior inspired by natural organisms are of great interest for investigating fundamental interfacial phenomena[1–4], as well as for addressing practical issues in the areas such as self-cleaning[5–7], fluid manipulation[8–10], icing reduction[11–13], drag reduction[14,15], and so on. In the past two decades, such a surface with coordinate multiphase wettability, in a highly selective and subtly triggered fashion, has drawn considerable attention in synthetic advanced superwetting membranes for applications ranging from oil/water separation to emulsion processing[16–19]. Although specific wetting and transport behaviors of water (surface energy, SE = 72.8 mJ m$^{-2}$) have been realized by tailoring surface chemistry and sophisticated microstructure[20,21], the precise management of organic liquids (OLs) remains a challenge because conventional superwetting membranes only exhibit single and homogeneous wettability for most OLs (the lower SEs, mainly below 35.0 mJ m$^{-2}$, and the smaller SE difference between each other). OL separation is regarded as the critical process in product purification, resource recovery, and microfluidic operation, which has become an exciting research frontier in recent years[22–24]. Current separation technologies mainly rely on thermal-based processes such as distillation, extraction, evaporation, forward, and reverse osmosis, which suffer from the disadvantages of multistep procedures, poor stability, costly device, and high-energy consumption[25–27]. As an emerging technology, the liquid purification by direct membrane filtration has considerable potential for improving environmental compatibility, reducing energy usage, and operation time compared with conventional thermal-based approaches but is rarely reported to deal with the extensive OL systems.

The ceaseless effort has been devoted to achieving the multiphase OL separation based on elaborate superwetting membranes governed by different wetting states (the wetted Wenzel and non-wetted Cassie–Baxter states). The primary strategy is assigned to regulate the SEs of the membranes between the intrinsic wetting thresholds of two immiscible OLs, resulting in the permeating of low SE liquid and the blocking of high SE liquid. Based on such a mechanism, there are some excellent works about the superwetting materials used for the OL separation including TiO$_2$ fibrous membrane modified with long-chain silanes[28], fluorosilane-doped polyvinylidene fluoride membrane[29], fluorinated CuO or Cu(OH)$_2$ nanoneedle mesh[30,31], fluorinated graphene/metal-organic framework composite membrane[32], and so forth. However, because of the uncertainty about the wetting threshold of OL pairs, it is relatively complicated to obtain such materials with targeted SE exhibiting opposite wettability for different liquids with tiny SE difference (below 5.0 mJ m$^{-2}$). The reported membranes have only routine one-way wetting mode based on the low SE and good solvent properties of the measured liquids, which are not suitable for practical processes, such as water removal in chemical reactions and solvent purification in organic systems. As for the tunable wetting modes, previous studies mainly focused on the water/oil system that the SE difference is as large as 60 mJ m$^{-2}$[33,34]. To the best of our knowledge, a demonstration of powerful wettability switch capabilities for various OLs is hitherto unreported. Hence, an intelligent membrane system capable of achieving full-SE-range controllable multiphase liquid separation has remained a distant prospect. It is inevitable to suffer from inherent limitations in modification operability, thresholds certainty, and separation universality.

Herein, we report a simple and scalable approach to obtain smart superwetting membranes with robust wettability switch capabilities for extensive OLs through the hierarchical orientation change of the two synergetic aggregation-induced layers with different SE components (Fig. 1a). Different from the mechanisms of the previously reported membranes, a physical parameter is proposed as surface energy component index (SECI, consisting of dispersive and polar parts) to describe the liquid interfacial and thermophysical properties. Liquid wetting or anti-wetting on the resulting surfaces exhibits a surprisingly straightforward connection with this physical parameter, suggesting that SECI can plausibly replace the generally recognized SE to decide the wetting behavior for each targeted OL on the surfaces. When the amphoteric fluorosurfactant layer points outward, the designed surface exhibits a significantly unique wetting tendency that liquid repellency increases with decreasing liquid SE. The joystick liquid can trigger the migration of the polyacrylate layer to alter the SE component contribution, realizing inversed wetting behaviors of various liquids. The tunable wetting and liquid-repellent properties of the membrane system allow it to be used in on-demand OL separation (at least 22 types of oily liquids system), in-situ operation of extraction or back extraction, wettability patterning, organic reagent purifying, and the composition determination (Fig. 1b). Through systematically analyzing the mechanism via the self-defined SECI, we reveal a general design principle of any surface with target wettability for full-SE-range OLs (a record-breaking SE difference as small as 0.3 mJ m$^{-2}$). This work not only provides a paradigm for the OL wetting mechanism to build an efficient bridge between arbitrary liquids and superwetting systems, but also unlock additional possibilities for the intelligent fluid-related systems owing to the simplicity, versatility, and low cost of the nanofabrication technique.

## Results

**Fabrication of the reconfigurable membranes.** Reconfigurable membranes were fabricated via a two-step "glue and paint" method that can construct versatile and stable superwetting interfaces based on the surface-embedding nanoparticles (Fig. 2a). A hydrogen bond self-assembly process using Capstone FS-50 (a short-chain amphoteric fluorosurfactant composed of partially fluorinated betaine) attached to commercial TiO$_2$ nanoparticles was first conducted to form the functional paint (Supplementary Fig. 1a, c). The transmission electron microscopy (TEM) images reveal that these resulting TiO$_2$ nanoparticles possess a uniform-size multi-granulous structure with a diameter of about 50 nm (Supplementary Fig. 2). The aforementioned result is consistent with the atomic force microscopy (AFM) images that the thickness of a single nanoparticle increases smoothly to ~50 nm (Fig. 2b). The element mapping of high-resolution TEM demonstrates the titanium, oxygen, nitrogen, and fluorine elements that are evenly distributed on the surfaces of TiO$_2$ nanoparticles, which is the side evidence of the successful modification of Capstone FS-50 (Supplementary Fig. 3). For further preparing the targeted membrane, the commercial polyacrylate adhesive (Supplementary Fig. 1b) was sprayed homogeneously onto the stainless steel mesh (SSM, the average pore size is ~50 μm, the thickness is ~80 μm, Supplementary Fig. 4) to form a glue layer bonding tightly the functional paint. Subsequently, the fluorosurfactant-assembled nanoparticles were anchored on the glue layer via the spray-coating approach, carefully regulating the balance of the spraying speed and solvent evaporation time. The as-prepared membrane (denoted as FS-SSM) displayed a rough coating surface with typical argenteous color (Supplementary Fig. 5). The scanning electron microscopy (SEM) images demonstrate the hierarchical textures composed of stacking micro-/nanoscale structures with multi-re-entrant curvatures attributed to the cooperative aggregates of adhesive nanoparticles and composite TiO$_2$ nanoparticles in mechanical interlocking mode (Fig. 2c, Supplementary Fig. 1d). Besides, the cross section of the FS-SSM is also observed as shown in Supplementary Fig. 6, indicating the aforementioned bilayer composite structures. Successful modification of fluorosurfactant-assembled nanoparticles is confirmed by X-ray diffraction that the

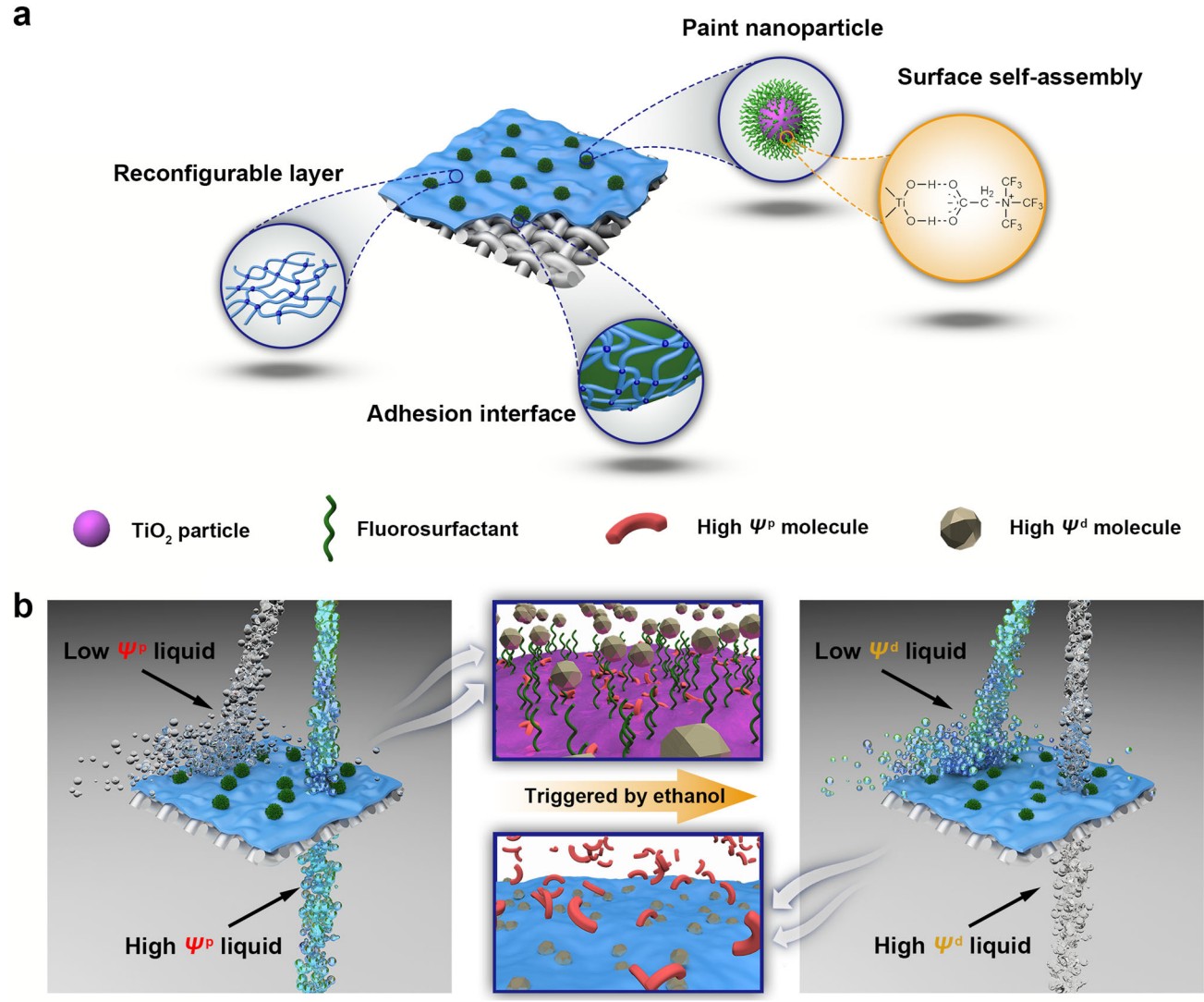

**Fig. 1 Schematic illustration of the reconfigurable membrane with tunable organic liquid (OL) wettability. a** The microstructure of the reconfigurable membrane. **b** On-demand OL separation procedures: the reconfigurable membrane demonstrates the lyophobicity for the low polar surface energy component index (SECI, $\Psi^p$) liquids and the lyophilicity for the high polar SECI liquids. After the joystick liquid (ethanol) triggers, the membrane will adopt an opposite wetting mode to accomplish OL separation based on the difference of dispersive SECI ($\Psi^d$).

diffraction peaks are well in accordance with the expected patterns of TiO₂ nanoparticles. (XRD, Fig. 2d). The X-ray photoelectron spectroscopy (XPS) and energy-dispersive X-ray spectroscopy (EDX) demonstrate the existence of fluorine-containing and nitrogen-containing functional groups on the surface of TiO₂ nanoparticles (Fig. 2e and Supplementary Fig. 7). Fourier transform infrared spectroscopy (FTIR) is conducted to examine the ingredients of the proposed layers and the hydrogen bond self-assembly process (Fig. 2f). Besides the stretching vibration of C=O and C–O–C groups of polyacrylate structure in the used adhesive at 1735, 1242, and 1157 cm⁻¹, several special peaks at 1625, 1325, and 1199 cm⁻¹ appear after modifying the paint nanoparticles. These special peaks are attributed to the COO⁻ and C–F groups originating from the fluorinated betaine structure of Capstone FS-50. It is worth noting that the red shift phenomenon is performed for the as-prepared membrane, which verifies the occurrence of the hydrogen bond self-assembly between fluorosurfactant and TiO₂ nanoparticles (Supplementary Fig. 8). Therefore, access to reconfigurable membranes was performed successfully according to the aforementioned characterization.

**Tunable wetting regularity of the reconfigurable membranes.** The wetting behaviors were investigated by measuring the contact angles of 15 commonly used liquids with a broad range of different polarities and total SE, categorized as polar protic liquids, polar aprotic liquids, and nonpolar liquids (Fig. 3a). The FS-SSM shows the superlyophilicity for polar protic liquids (water, formamide, and ethylene glycol), while anti-wetting property for nonpolar liquids (toluene, cyclohexane, dodecane, dichloromethane (DCM), dichloroethane, tetrachloromethane, diiodomethane), as observed by combining the photographs of the representative liquid droplets deposited on the membrane (Fig. 3d and Supplementary Movie 1). Besides, it can be seen that the membrane is particularly distinctive in the interaction with polar aprotic liquids. For example, N,N-dimethylformamide embodies superlyophilicity on the membrane surface while the opposite wettability for nitromethane, though both have incredibly similar SE values (36.5 and 36.8 mJ m⁻², respectively). The influence of the substrate on the aforementioned unique wettability can be ignored as shown in Supplementary Fig. 9. The effect of the fabricated surface hierarchy to enhance the

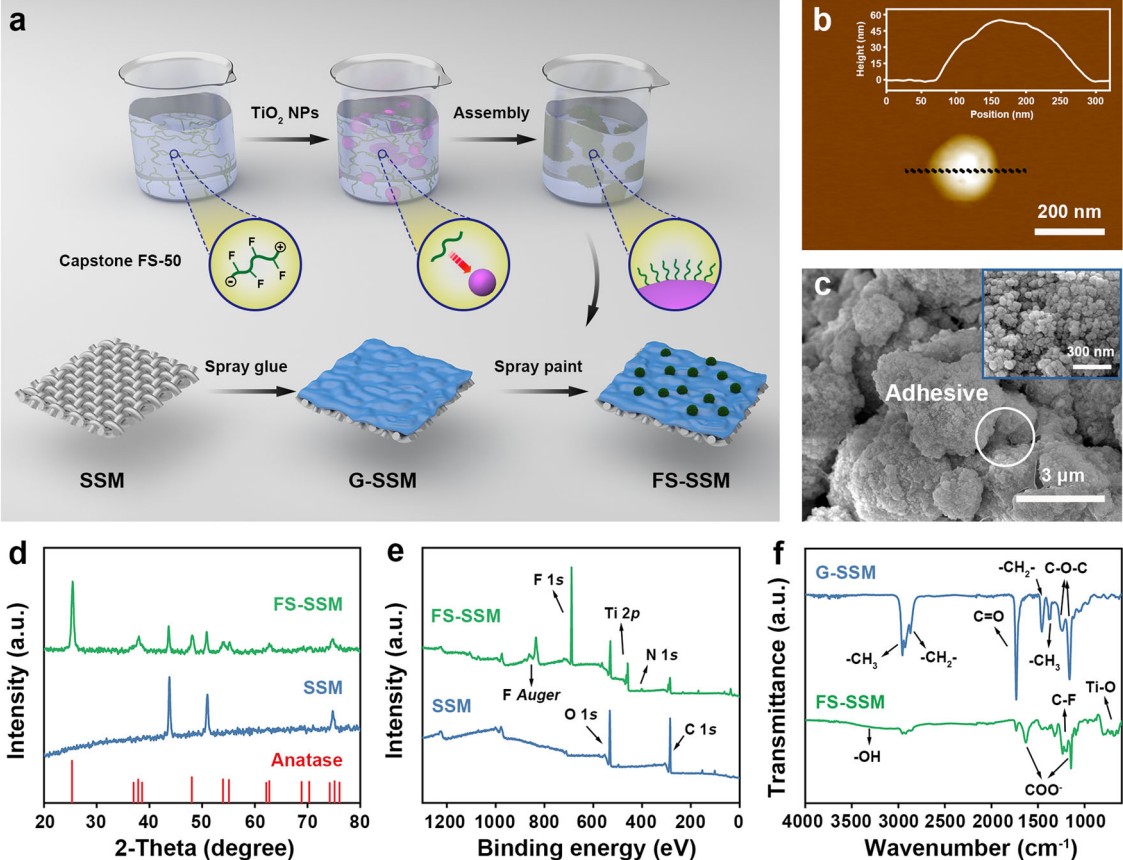

**Fig. 2 Fabrication process and characterizations of reconfigurable membranes. a** Schematic diagram of the preparation procedures (G-SSM represents the polyacrylate glue-coated SSM). **b** AFM images with height profile along the dotted line. **c** SEM images of the as-prepared membrane showing the hierarchical microstructures. **d** XRD pattern of FS-SSM and untreated SSM substrate compared with the respective standard patterns of $TiO_2$ anatase. **e** XPS wide spectra of FS-SSM and SSM, where "F" and "N" refer to the Capstone FS-50, and "Ti" refers to $TiO_2$. **f** FTIR spectra of the membranes in different preparation stages.

lyophobicity and low adhesive performance is further investigated based on a recursive form of the Cassie–Baxter relation, indicating the best-performing non-wetting status is conducted by the co-spray-coating method (four levels of hierarchy, Supplementary Figs. 10–12, Supplementary Note S1, Supplementary Movie 2).

Unexpectedly, the as-prepared membrane after certain specific liquids processing (methanol and ethanol, strong polar solvents that are miscible with most OLs) exhibits opposite wetting behaviors that could coincide with conventional superwetting mode (Fig. 3d). Whereas the membrane still possesses high stability for other wetting liquids due to the hydrogen bond and mechanical interlocking interactions of the proposed layers (Supplementary Fig. 13, Supplementary Table 1). Supplementary Fig. 14 ascertains the dynamic wettability transition process through the measured contact angles of water and ethylene glycol on the membrane as a function of the ethanol immersion time. As the treatment time increased, water repellency is gradually enhanced, and the surface reaches a stable superhydrophobicity state in a brief period. Besides, we can find that although the surface has the intensive EG repellency in the preliminary stage, the corresponding contact angle shows an apparent decrease at about 210 s. This phenomenon inspires us to choose the ethanol as an excellent trigger to achieve the surface property switching (the FS-SSM treated in the ethanol for 3 min as an example called PR-SSM for subsequent study). Figure 3b shows PR-SSM has almost inverse wetting behaviors for the measured OLs compared with FS-SSM, inhibiting the wetting of polar protic liquids and polar aprotic liquids, whereas substantially lyophilicity to

nonpolar liquids except the diiodomethane (Supplementary Fig. 15, Supplementary Movies 3 and 4). Although previous reported lyophobic surfaces possess excellent liquid repellency, they all show uniform wettability for OLs or wetting tendency that conforms to the rule of classical solid–liquid total SE comparison. As shown in Fig. 3f, the inverted and different wettability for OLs in our case is rare that represents a sharp departure from previous works[28–31,35,36].

**Mechanism of the wetting behaviors and transition.** The initial wetting behaviors are quite different from the well-known phenomena because such a surface shows the non-linear and counterintuitive relationship between the liquid contact angle and the corresponding total SE (Supplementary Fig. 16a). The conventional wetting mechanisms cannot also explain the wettability switch process because previous reports only focused on the wettability transition for water rather than OLs. We proposed a mechanism from the perspective of the interfacial thermodynamics to explore the in-depth investigation of the above phenomena. The interaction between solid and liquid results from various types of well-known molecular forces, such as London dispersion forces and hydrogen bonding[37,38], implying that the comprehensive evaluation of the wetting behaviors based on different SE components is required (Supplementary Note 2 gives a detailed discussion). The results suggest that the arbitrary liquid's wetting behavior is decided through the component contribution accompanied by the total SE, rather than the single impact previously proposed. A well-designed parameter is further

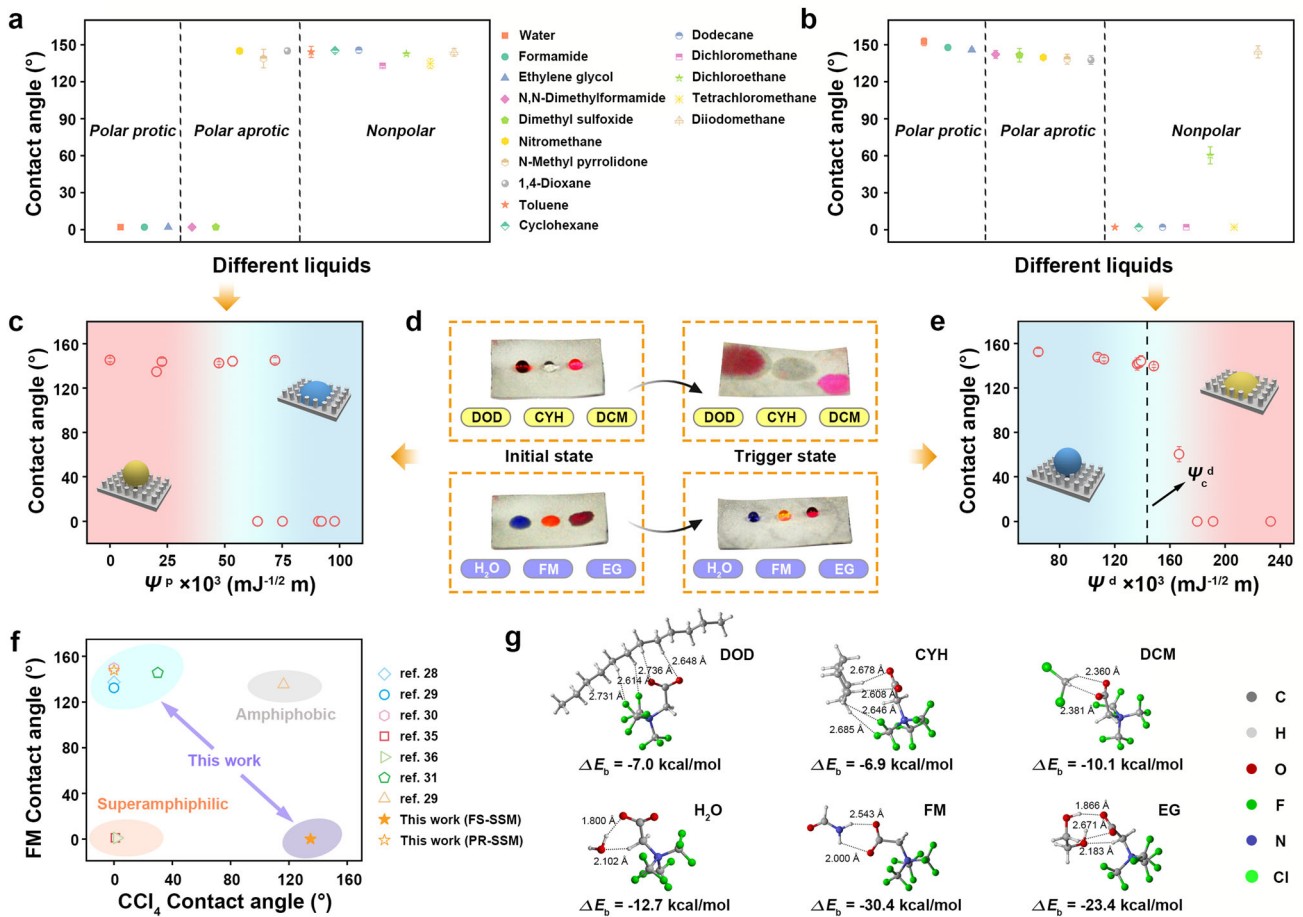

**Fig. 3 Wetting transition behaviors of the membranes. a** Contact angles on the FS-SSM for various OLs including polar protic liquids, polar aprotic liquids, and nonpolar liquids. **b** Contact angles on the PR-SSM for above OLs. **c** Dependence of contact angle on the $\Psi^p$ of the corresponding liquids (FS-SSM, each data is explained in Supplementary Table 4). **d** Left section: Digital photographs of different liquid droplets on the FS-SSM including dodecane (DOD), cyclohexane (CYH), and dichloromethane (DCM), water, formamide (FM), and ethylene glycol (EG); Right section: Digital photographs of the corresponding liquid droplets on the surface after the trigger of ethanol. **e** Dependence of contact angle on the $\Psi^d$ of the corresponding liquids (PR-SSM, each data is explained in Supplementary Table 5). **f** Comparison of wetting performance of our membranes with some other notable reported membranes in the literature, taking formamide (FM) and tetrachloromethane (CCl₄) as the research models. **g** Optimized geometries of model OL molecules binding with the amphoteric fluorosurfactant molecule by the density functional theory (DFT) calculations and the corresponding binding energy ($\Delta E_b$).

developed to denote versatilely the real OL interfacial property, which is intituled as SECI ($\Psi$) comprising dispersive part and polar part:

$$\Psi = \sqrt{\frac{\varphi_l}{\gamma_l^d + \gamma_l^p}}, \tag{1}$$

where $\gamma_l$ represents the surface energy of the liquid, the superscripts d and p refer to the dispersion force and polar force components, $\varphi_l$ is the proportion of dispersive part or polar part in total liquid SE. The theoretical wetting relation can be expressed as

$$\cos\theta = 2\sqrt{\gamma_s^d}\Psi^d + 2\sqrt{\gamma_s^p}\Psi^p - 1, \tag{2}$$

where $\theta$ is the equilibrium contact angle of the liquid on the flat solid surface, $\gamma_s$ represents the surface energy of solid, $\Psi^d$ and $\Psi^p$ are the dispersive and polar SECI of the liquid, respectively. Notably, our membrane at the initial state adopts in low dispersion component (fluoric groups) and high polar component (quaternary ammonium salt with carboxylate) that represents one extreme case ($\gamma_s^p \gg \gamma_s^d$) of Eq. (2). The corresponding configuration of polar SE and dispersive SE of FS-SSM was measured, that the $\gamma_s^p$ was as high as 77.35 mJ m⁻², whereas $\gamma_s^d$ was as low

as 3.56 mJ m⁻², as anticipated (Supplementary Fig. 17, Supplementary Note 3, Supplementary Table 2). Supplementary Fig. 16b illustrates the theoretical contact angle of the arbitrary liquid on the FS-SSM estimated by $\gamma_l^p$ and $\gamma_l^d$ according to the OWRK model[39]. As signified in this SECI-driven theory, $\Psi^p$ and $\Psi^d$ are two more reasonable parameters to monitor the wetting behaviors. With respect to this case, we establish a simplified model to describe the wetting properties of the FS-SSM that the dispersive SE of the membrane is negligible and the total SE is equal to the contribution of polar SE. Equation (2) can be described as

$$\cos\theta = 2\sqrt{\gamma_s^p}\Psi^p - 1, \tag{3}$$

The intrinsic $\Psi^p$ of the used liquids were calculated accurately by Eq. (2) as summarized in Supplementary Table 3. As manifested in Fig. 3c, our membrane exhibits the ideal data correlation and the similar wetting tendency with the theoretical prediction (the affinity for the liquids with high $\Psi^p$, in striking contrast, the liquid-repellent property for the counterparts with low $\Psi^p$). Density functional theory (DFT) was employed to quantitatively verify the above-proposed hypothesis (Fig. 3g and Supplementary Note 4). Hence, the correctness of our empirical

hypothesis has been validated, whether experimentally or computationally.

The mechanism underpinning the unusual wetting transition phenomenon can be attributed to the adhesive layer that serves as both the adhesive and tunable wettability trigger for the reconfigurable membrane. During the ethanol-assisted process, the crosslinking polyacrylate network can partially swell and spontaneously migrate to the surface of paint nanoparticles driven by the capillary force, leading to the rearrangement of these particles in response to the adhesive flowing[40,41]. Such a mechanism is supported by the systematical characterization of surface morphology and chemical composition that the corresponding signal of the polyacrylate layer can be strongly detected on the PR-SSM surface after the joystick liquid treatment (Supplementary Fig. 18). The dynamic migrating process is further demonstrated by utilizing a laser confocal microscope. The changing visible blue fluorescence of the polyacrylate layer can be clearly observed in the confocal images of the as-prepared membrane via joystick liquid treatment for different time (Supplementary Figs. 19 and 20a–g). It can be seen that the fluorescence intensity gradually increases with treatment time, suggesting the dynamic migrating process of the polyacrylate layer triggered by the joystick liquid (Supplementary Fig. 20h). The adhesive-induced particle rearrangement endues the surface with different configurations of SE components. On the one hand, the migration of polyacrylate molecules is inclined to reduce the outermost layer's polar SE due to the mainly dispersive component in adhesive SE. On the other hand, that is a common occurrence that the outermost layer's dispersive SE would be enhanced by replacing the F element with other elements in adhesive. According to Eq. (2), the balance of both gives rise to the appearance of the maximal contact angle value of certain liquids such as ethylene glycol during the treatment process. Therefore, the appropriate reconstruction extent of the surface is decisive in obtaining the required wetting mode. From the SECI-driven theory perspective, we can determine and explain the all wetting phenomenon of the chosen PR-SSM. The polar SE and dispersive SE values of PR-SSM were calculated as 24.57 and 2.22 mJ m$^{-2}$, respectively (Supplementary Table 2). Equation (2) can be simplified as an approximation that is suitable in case of the above condition ($\gamma_s^p << \gamma_s^d$):

$$\cos \theta = 2\sqrt{\gamma_s^d \Psi^d} - 1. \qquad (4)$$

Hence, the liquid affinity of this type of SECI-driven membrane should display a positive $\Psi^d$-response hallmark. As expected, we find that the contact angles of OLs on the PR-SSM are inversely proportional to the respective $\Psi^d$, which unfolds the similar variation tendency as predicted by Eq. (4) (Fig. 3e). Thereinto, $\Psi_c^d$ is defined as the critical dispersive SECI for liquids on the targeted surface, meaning the distinction between lyophobicity and lyophilicity. The theoretical $\Psi_c^d$ of the solid surface can be empirically estimated when the intrinsic contact angle threshold for liquids is about 65°, resulting that the corresponding value of PR-SSM is 0.1435 mJ$^{-1/2}$ m (derived from Eq. (2))[42]. By comparing the $\Psi^d$ value of each liquid with this critical point, it seems that the wetting behaviors can be well explained even for the exceptional case in nonpolar liquids (diiodomethane, $\Psi^d = 0.1390$ mJ$^{-1/2}$ m).

**On-demand OL separation.** Behaved as the aforementioned SECI-based protocol, our nanoparticle-embedded membranes with tunable wettability is anticipated to effectively and controllably separate a wide range of immiscible OL pairs through direct gravity filtration. The proof-of-concept experiments were first acquired using apparatus equipped with FS-SSM to release

the high $\Psi^p$ liquids but hinder the low $\Psi^p$ liquids (Fig. 4a and Supplementary Movie 5). To evaluate the separation capacity of the membrane, nine types of OL model pairs with various SE differences (containing oil–water system and oily liquids system, Supplementary Fig. 21) were performed the two-phase separation process. The separation efficiencies of the as-prepared membrane all reach above 98.0% for the measured liquid mixtures (Fig. 4b). Besides, the permeation flux of the membrane possesses a considerable variation due to the viscosity difference of the separated liquids (Fig. 4c). The membrane performs an average flux above 3000 L m$^{-2}$ h$^{-1}$ for the mixtures containing formamide or water under gravity, while the flux would decrease to about 1000 L m$^{-2}$ h$^{-1}$ for high-viscosity liquid (ethylene glycol). The increased viscosity has no effect on the separation efficiency except for a decrease in the flux of the separated liquids. Such a separation process can be repeated at least 35 cycles without apparent separation efficiency decline and permeation flux loss, indicating the excellent long-term reusability (Supplementary Fig. 22a). In such separation process, the Laplace pressure ($\Delta P$) is crucial to the separation as described in the Young–Laplace equation:

$$\Delta P = \frac{4\gamma_l \cos \theta_a}{D_{pore}} = \frac{4\left[2\left(\sqrt{\gamma_s^d \gamma_l^d} + \sqrt{\gamma_s^p \gamma_l^p}\right) - \gamma_l^d - \gamma_l^p\right]}{D_{pore}}, \qquad (5)$$

where $D_{pore}$ represents the average pore diameter of the membrane calculated as 4.84 μm (Supplementary Fig. 23, Supplementary Note 5). Figure 4h reveals the dependency of $\Delta P$ and the corresponding liquid SE components at the obtained $\gamma_s^d$ and $\gamma_s^p$ of the FS-SSM. It is well-known that the critical Laplace pressure ($\Delta P_c$, the red tilting plane) for droplets to permeate is associated with the intrinsic wetting threshold for liquids. In our assumption, $\theta = 65°$ is considered as intrinsic contact angle threshold for liquids rather than the previous mathematical threshold of $\theta = 90°$. Hence, the liquid droplets can permeate the membrane in the case of $\Delta P > \Delta P_c$ while are blocked in the opposite case of $\Delta P < \Delta P_c$.

Similarly, in light of the dependence of contact angles on $\Psi^d$, OL separation arises from the PR-SSM actuated by the reverse wetting mode against the case of FS-SSM, in which high $\Psi^d$ liquid selectively permeates while heterogeneous low $\Psi^d$ liquid is retained (Fig. 4d and Supplementary Movie 6). The separation efficiencies for all the nine typical OL pairs are above 97%. The membrane exhibits high liquid fluxes (above 4000 L m$^{-2}$ h$^{-1}$) and cyclic performance (over 35 cycles) during the separation process (Fig. 4e, f, and Supplementary Fig. 22b). Aiming to facilitate the smart collection of OL mixtures, a self-adaptive fluidic separation device with two outlets is designed by integrating FS-SSM and PR-SSM, which serves as the gates of high $\Psi^p$ liquid and high $\Psi^d$ liquid, respectively (Fig. 4g and Supplementary Movie 7). The continuous, automatic, on-demand separation of at least 22 types of other complicated liquid mixtures with a tiny total SE difference (such as ethylene glycol/diiodomethane, dimethyl sulfoxide/cyclohexane, and dimethylformamide/dodecane) has been realized, indicating that the membrane is a promising material for high-efficiency OL separation and targeted liquid gating (Supplementary Table 6). These results demonstrate that the membrane system is a good candidate for smart and high-efficiency liquid separation. Furthermore, our membrane can overcome the weak inherent robustness of superwetting membranes based on the sophisticated micro/nanostructure, with the nanoparticles design to provide liquid repellency and an adhesive layer design to provide durability (Supplementary Fig. 24).

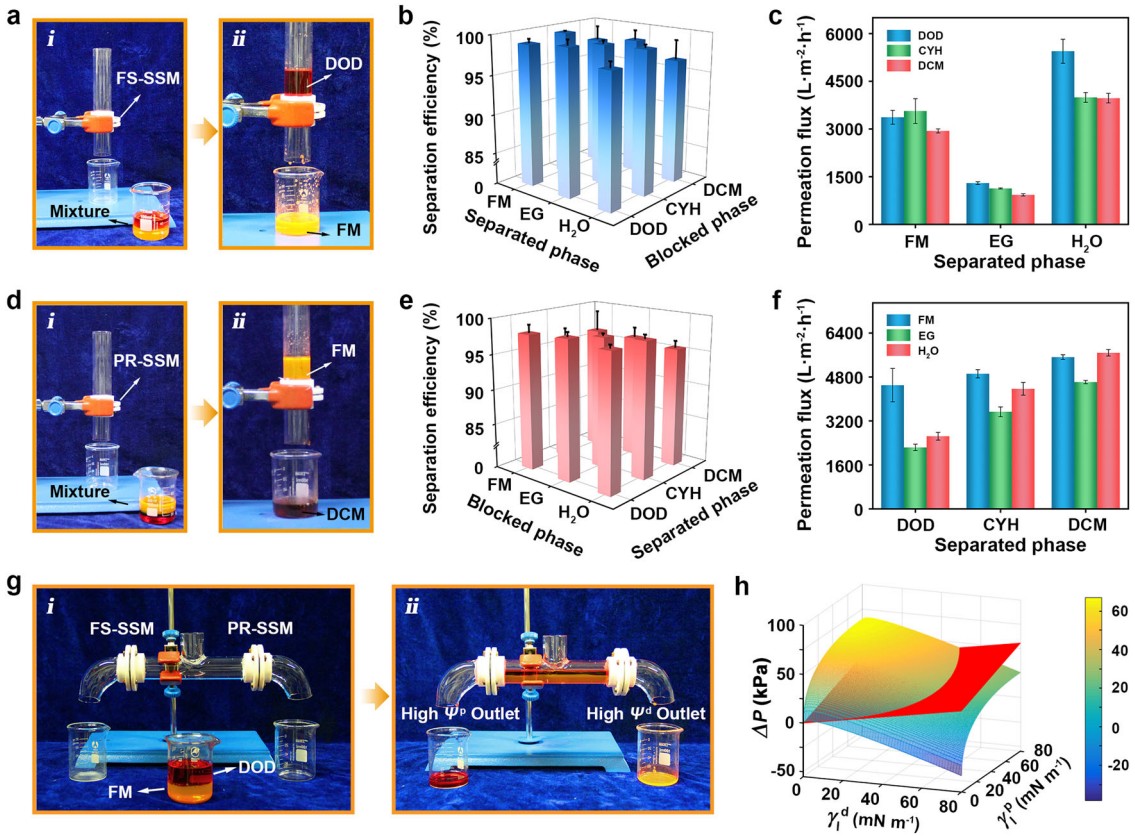

**Fig. 4 On-demand OL separation of the membrane system. a** Separation process of immiscible FM/DOD mixture that FM passes through FS-SSM but DOD is remained (from i to ii). **b** Separation efficiencies for different OL pairs of FS-SSM. **c** The permeation flux of the FS-SSM for the above liquid mixtures. **d** Inverse separation process of the FM/DCM mixture using PR-SSM, in which DCM passes through but FM is remained (from i to ii). **e** Separation efficiencies and **f** the corresponding permeation flux for different OL pairs of PR-SSM. **g** Smart separation of FM/DOD mixture through a self-adaptive fluidic separation device with two outlets. FM (high $\Psi^p$ liquid) has a strong permeability to the FS-SSM gate, while the other PR-SSM gate provides passable channels for DOD which is ascribed to its high $\Psi^d$. **h** $\Delta P$ as a function of $\gamma_l^d$ and $\gamma_l^p$ for impacting the separation of OL mixtures for FS-SSM. The red tilting plane represents critical Laplace pressure ($\Delta P_c$).

**In situ dual-type extraction and other applications**. Liquid-phase extraction has been widely applied in the environmental and industrial fields of separation and analysis. However, traditional extraction routes are limited by the inherent drawbacks, involving complex procedures, tedious operating time, low back extraction efficiency, costly device, and single extraction type[43–46]. Based on the tunable wettability of the reconfigurable membranes, we carry out in situ real-time operations of extraction and back extraction, which offer an excellent choice to realize a variable, rapid, efficient, selective removal of densely packed extraction agents (Fig. 5a). A model of in situ extraction device was designed to conduct the real-time operation by assembling the FS-SSM vertically without compromising the densities of the possessed liquids. The initial membrane's ability to function as a back extraction system was first accessed by taking the critical phenol back extraction process in the actual chemical industry as a typical model. DCM solution of phenol was considered as the stock solution to be processed, and 10 wt% NaOH aqueous solution was selected as a back extraction agent for the extraction of the phenol solute. Our FS-SSM exhibits different wetting behaviors for the chosen liquids (DCM-repellency and water-affinity in Supplementary Fig. 25) due to the polar SECI discrepancy. The corresponding in situ back extraction route was deployed with constantly feeding of the fresh back extraction agent into the system, which worked in the mode of removing phenol-rich back extraction agent while repelling the stock solution simultaneously (Fig. 5b and Supplementary Movie 8). As

controls, we gave insight into the traditional back extraction using the separating funnel, which entailed more complicated procedures comprising the so-called drastic mix, gravity settlement, and static separation (Supplementary Fig. 26, Supplementary Movie 9).

The back extraction system volume was set as 75 mL with a volume ratio of back extraction agent and DCM (50:25). By investigating the real-time concentration of phenol in the back extraction agent (Fig. 4c, with the standard curves of phenol in NaOH solution of Supplementary Fig. 27a), it is obvious that the back extraction agent enriches the phenol efficiently and continuously with arriving an optimized state at circa 3 min. In comparison, the traditional route has not still reached a steady state in the same operation time as the former (Fig. 5d). Figure 5e reveals more excellent efficiency (~82%) of in situ FS-SSM back extraction than the traditional method (~75%). Volume recovery of the back extraction agent by in situ strategy (~98%) is also higher than that conducted by a traditional route (~95%). In addition, the DCM content in the collected back extraction agent (~30 ppm) of our membrane system was reduced by 70% compared with traditional back extraction (~100 ppm). Thus, a superior integration of solute back extraction and liquid separation was successfully realized based on the in situ back extraction system containing the FS-SSM.

The membrane after wettability transition (PR-SSM) is further available for the in-situ extraction operation. The difference from the above operation is that aqueous phenol solution and DCM act

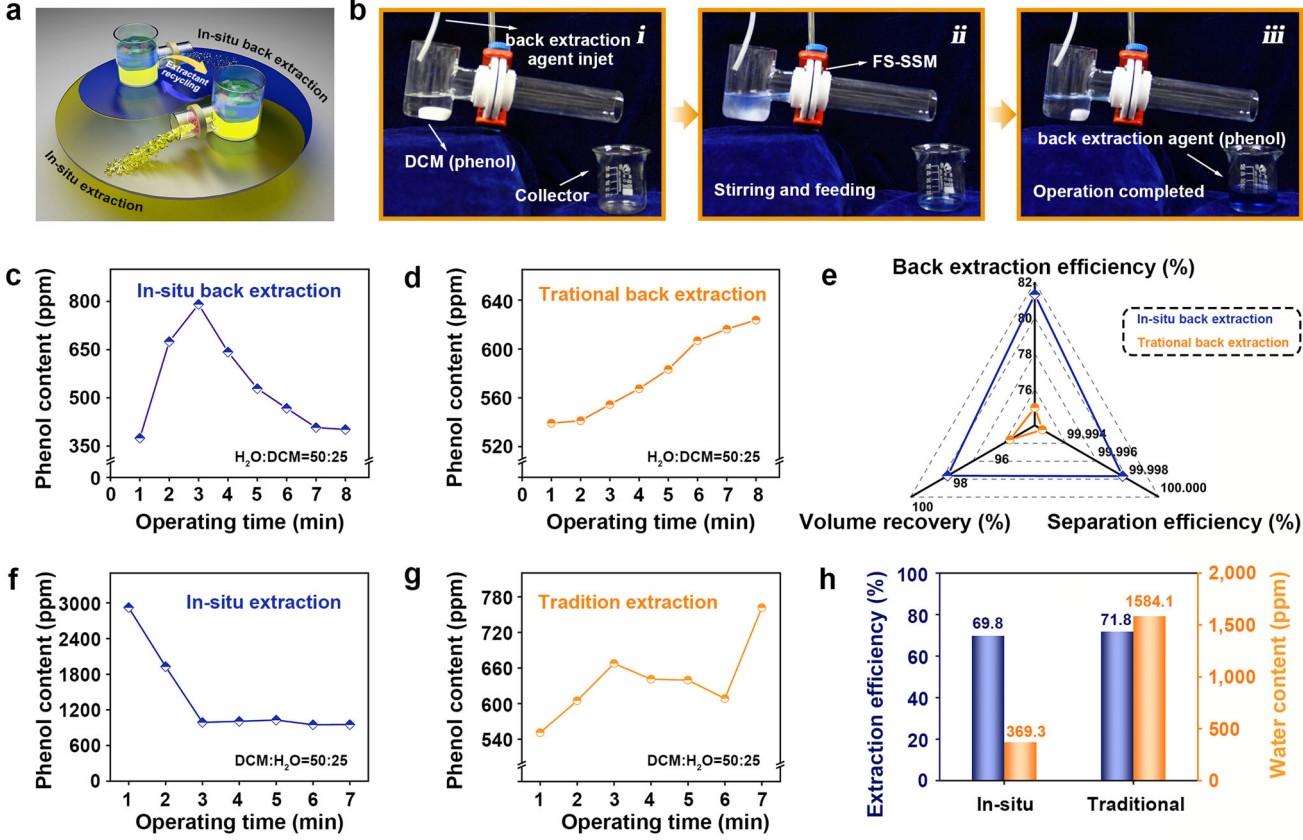

**Fig. 5 In situ extraction/back extraction using the reconfigurable membrane. a** Schematic illustration of the extraction-back extraction coupling and recycling process. **b** Digital photographs of the in situ back extraction process based on FS-SSM (i–iii). Real-time concentration of phenol in the back extraction agent phase during **c** the in situ back extraction process and **d** the traditional route. **e** Evaluation to back extraction performances of two methods from three different evaluation perspectives containing back extraction efficiency, volume recovery of the back extraction agent, and separation efficiency. Real-time concentration of phenol in the extraction agent phase during **f** the in situ extraction process and **g** the traditional route. **h** Extraction efficiency and water content in the collected extraction agent of the in situ extraction and traditional methods.

as the stock solution and extraction agent, respectively. The PR-SSM endows wetting differences towards the pending two-phase system, exhibiting superlyophilicity to the extraction agent and high-lyophobicity to the water phase (Supplementary Fig. 28). Such wetting behaviors enable us to realize the extraction agent's in-situ removal contained phenol from aqueous solution (Supplementary Fig. 29, Supplementary Movie 10). For a certain volume (75 mL) of the extraction system (the volume ratio of extraction agent and aqueous phenol solution is 50:25), Fig. 5f and g denote the real-time concentration evolution of phenol in the extraction agent during the in-situ extraction process and traditional operation using the standard curves of phenol in DCM (Supplementary Fig. 27b). It can be seen that the in-situ process reaches a steady state at about 3 min, while the traditional method remains a fluctuant state with a slight change. The extraction efficiency of the PR-SSM is comparable to that of the traditional way, but more complicated procedures and sharply higher water content in the extraction agent are required for the traditional operation (Fig. 5h; Supplementary Fig. 30, Supplementary Movie 11). Therefore, our in situ membrane system is more competent for achieving extraction in a facile, rapid, and efficient mode than traditional routes. The proposed strategy as a versatile platform can also be used on various substrates and of great significance to a wide range of applications. For example, "THU" patterns with various self-defined colors could be generated on the stainless steel substrate by utilizing the distinct OL wettability (Supplementary Fig. 31). Melamine sponge dip-coated with the

as-prepared paint can absorb water from OLs such as toluene without any contamination, realizing highly efficient organic reagent or chemical reaction system purifying (Supplementary Fig. 32). We also design an intelligent sensor combined with the distinct SECI-driven wettability to determinate a binary miscible mixture's composition (Supplementary Fig. 33, Supplementary Note 6).

## Discussion

In summary, a self-defined physical parameter is developed that can classify full-SE-range liquids and consider the comprehensive effect of SE and its different components using a more reasonable way. Based on the proposed SECI theory, we report superwetting membranes with tunable OL wettability, obviating the previous sophisticated control of the membrane's SE. Relying on changeable SE components, the fabricated membranes preferentially capture and interact with the corresponding SECI-dominated liquids, favoring the wide range of controllability and high-efficiency into a self-propelled fluidic separation system. Additionally, in situ extraction and back extraction operations using the system are superior to conventional counterparts, such as enhancing enrichment efficiency, reducing heterogeneous residues, increasing recovery, and simplifying procedures. Taking into account the operational simplicity and wettability difference, by enabling the generation of the paint in various substrates, the fabricated materials greatly expand the potential application in

patterning, purifying, and detection fields. Unlike other reported superwetting membranes, from the perspective of the new-developed interfacial theory, the distinct wettability performance, the controllable liquid–liquid separation, and the expansibility in applications, our study updates the understanding of wetting behaviors of different OLs, which should offer more elevating options for the viability of microscale reactor operation, lab-on-a-chip settings, and beyond.

## Methods

**Fabrication of the reconfigurable membranes**. A spray-coating suspension was first prepared by simply mixing amphoteric fluorosurfactant (Capstone™ FS-50, Dupont) with nanoparticles. First, 5.0 mL Capstone FS-50 was dissolved in 100.0 mL ethanol solution, and the whole mixture was strongly stirred for 1 h at room temperature in a sealed condition. Afterward, 3.00 g of TiO₂ nanoparticles (anatase, Aladdin Industrial Inc.) were dispersed homogeneously in the above solution with continuous stirring for another 2 h to prepare a homogeneous opalescent suspension. In the following experiments, aiming to remove the dirt and oxides on the substrate surface, the stainless steel meshes (4 × 4 cm, 400 mesh size) were primarily rinsed by the ultrasonic cleaner in the mixed solution of deionized water, ethanol, and acetone for 0.5 h. Subsequently, the Spray-Mount™ Super 75 (3M™) adhesive was loaded homogeneously onto the SSM substrate, followed by the spraying coating of the as-prepared suspension using a spray-gun for 5 min. The air pressure for spraying was maintained at 0.1 Mpa, the distance between the spray-gun and the substrates was ~15 cm, and the spraying speed was ~0.5 cm s⁻¹. As for the fabrication of PR-SSM, the above membrane was immersed in absolute ethanol for 3 min and dried under ambient conditions for further operation.

**Fabrication of the flat intrinsic samples**. Silicon wafers used in this study were cut into 2 × 2 cm before use. Then Capstone FS-50 was applied on a silicon wafer by spin-coating as the flat intrinsic state of FS-SSM. Similarly, as for the intrinsic state of PR-SSM, adhesive and Capstone FS-50 was successively spin-coated on the silicon wafer, followed by the treatment of ethanol for a certain time. All flat samples were dried on the thermal platform.

**Immiscible OL mixtures separation test**. The evaluation of the separation performance of immiscible OL mixtures was conducted using a dead-end filtration device. A piece of the as-prepared membrane was fixed between two Teflon fixtures connected to two glass funnels with an effective separation area of 3.80 cm². The OL mixtures (50.0 mL in a volume ratio of 1:1) were directly poured into the device through the glass tube. The OL separation efficiency was measured three times in one separation cycle, calculated in terms of the liquid rejection coefficient ($\eta_m$) from the following equation:

$$\eta_m = \left(1 - \frac{m_t}{m_0}\right) \times 100\%,\qquad(6)$$

where $m_0$ and $m_t$ represent the mass of permeable liquid before and after the separation. To eliminate the influence of the membrane absorption, the separation efficiencies were calculated after the initial separation[47–49]. The permeation flux of the membrane was determined by calculating the volume of the separated phase in unit time using the following equation:

$$F = \frac{V}{At},\qquad(7)$$

where $F$ is the flux of the membrane, $V$ is the volume of the separated phase, $A$ is the effective area of the membrane (3.80 cm²), and $t$ is the testing time. Each data point was the average value of three parallel experiments. Besides, smart OL separation proceeded through the self-adaptive device assembled with two channels. FS-SSM and PR-SSM were fixed with polytetrafluoroethylene flanges and installed on both sides of the "inverted T shape" unit. During the process of OL mixture pouring, different organic phases were collected from two outlets.

**Continuous in situ extraction and back extraction test**. The as-prepared membrane was assembled between a pair of flanges in the designed in-situ device in a vertically placed way. The DCM solution and the aqueous solution of phenol were prepared at a concentration of 500 ppm. Then 25.0 mL phenol solution was added into the left part of the in situ extraction device once under stirring (400 r/min). Fresh extraction agent or back extraction agent was dropwise added to (8.0 mL/min) into the device for enriching the phenol solute. The final filtrates were entirely collected in the collector. A traditional operation for comparison was manipulated by applying similar parameters with in situ extraction. For the evaluation of extraction or back extraction performance, the liquid system volume was kept at 75.0 mL in which the volume of extraction agent/back extraction agent occupies two-thirds of the total volume. Supplementary Note 7 offers the corresponding description of evaluation indexes.

**Patterning, organic reagent purifying, and composition determination**. The stainless steel substrate was designed as a "THU" pattern and spray-coated with the polyacrylate adhesive and the as-prepared paint. Combining with the trigger of the ethanol, we can obtain different patterns with specific colors by the target liquid wetting. The porous melamine sponge was dip-coated in the above suspension for 10 min, then taken out by a pair of tweezers and dried immediately. This process was repeated five times, followed by drying at 60 °C to get the modified sponge absorbent. Toluene/water mixture was used as an example to demonstrate the organic reagent purifying process. As for the composition sensor, the device is first customized of polytetrafluoroethylene (PTFE) with a set of parallel grooves. The polyacrylate adhesive and resulting paint were only spray-coated in the designed grooves through covering the outside surface with the polyimide tapes. A series of acetonitrile/water mixtures (30 μL) was dropped into the grooves to exhibit different wetting lengths that indicate gradient composition ratios.

**Characterization**. The microstructure of the as-prepared TiO₂ nanoparticles was characterized by high-resolution transmission electron microscopy (HR-TEM, JEM 2100F, JEOL, Japan) and atomic force microscope (AFM, SPM-9600 series, Shimadzu, Japan). The surface morphology of the membranes was recorded by field mission scanning electron microscope (FESEM, SU-8010, Hitachi Limited, Japan). The chemical composition of the samples was determined with the means of X-ray diffraction spectroscopy (XRD, Bruker D8 Advance, Bruker-AXS, Germany), XPS (PHI Quantera SXM, ULVAC-PHI, Japan), and Fourier transform infrared spectroscopy (FTIR, V70, Bruker, Germany). The fluorescence spectra were measured by a fluorescent photometer (RF-2000, SHIMADZU, Japan). Confocal images were collected by a laser confocal microscopic system (LSM-780, Zeiss, Germany). A PerkinElmer Lambda 750 UV spectrometer (United Kingdom) was used to test the phenol concentration. The contact angles were probed with a contact angle measurement machine (OCA 15 machine, Data-Physics, Germany). Moreover, the final contact angle of each sample was calculated by measuring three contact angles of different positions and taking the average. In the liquid dropping tests, the motion of the test droplets was captured with the help of a digital still high-speed camera (FASTCAM SA-Z, PHOTRON, Japan) of 10,000 fps.

**Computational details**. In order to gain further insight into the intermolecular interactions, we carried out DFT calculations. All the structures of the global minima were optimized at the ωB97X-D/6-31G(d,p) level of theory[50–52]. The ωB97X-D functional was chosen due to its accurate description of thermochemistry, kinetics, and non-covalent interactions. The frequencies were calculated to verify the global minima and provide the zero-point energy (ZPE) corrections. All the theoretical calculations were carried out using Gaussian09 package[53]. The binding energies of the model complexes were calculated according to Eq. (8):

$$\Delta E_b = E_{\text{Capstone FS}-50/\text{Liquid}} - \left(E_{\text{Capstone FS}-50} + E_{\text{Liquid}}\right).\qquad(8)$$

## Data availability
Supplementary Figs. 2, 8, 9, 12, 13, 14, 16a, 18c–e, 19, 20h, 21, 22, 23, 24b, 27, 33d are provided as a Source Data file. The data that support the findings of this study are available from the corresponding author upon reasonable request. Source data are provided with this paper.

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

## Acknowledgements

This work is supported by the National Natural Science Foundation (51173099, 21788102). The authors are grateful for the colleagues at Tsinghua University for experiment support, Prof. Liangti Qu and Dr. Haiyan Wang for the spraying instrumentation guidance, Dr. Fuwei Sun, Dr. Wei Hang, and Dr. Guoqiang Liu for useful discussions.

## Author contributions

X.L. and L.F. conceived the ideas of smart OL gating membranes presented. L.F. guided the work. X.L. planned and executed all of the experiments, with the help of R.Q. and W.Z. on the data analysis. J.L. and H.H. performed the density functional theory calculations. X.L. and L.F. wrote the paper and interpreted the results with comments from Y.L., H.Z., and Y.W.

## Competing interests

The authors declare no competing interests.
