## [Peer Review File · Nature Communications]

REVIEWER COMMENTS

Reviewer #1 (Remarks to the Author):

C1: In Fig. 1a, the high Ψ_p molecules are expressed in the form of a rod. In fact, most of the polar molecules are bent like water molecules rather than the rod shape. It would be easier to understand if you express it with a curved shape and add a color to express the polarity.

C2: In Fig. 1, both the fluorosurfactant and the polyacrylate molecule of the adhesion layer are represented in green, which is confused. It would be good to modify the color so that they can be distinguished.

C3: In of Supplementary Figure 18, it is recommended to perform the long-term reusability experiment more than 30 cycles because it usually shows a phenomenon of poor durability from 10 cycles or more.

C4: Separation efficiency is generally lowest in first cycle and high in other cycles, but the separation efficiency in Supplementary Figure 18 shows a different trend. Please explain why.

C5: In order to reinforce the originality of the paper, additional experiments are required to prove the mechanism or principle for the change of the membrane properties by joystick liquid. This is because papers that change the properties of membranes using joystick liquids have already been reported.

Reviewer #2 (Remarks to the Author):

In this manuscript, the authors developed a nanoparticle-embedded gating membrane composed of different reconfigurable layers, which can realize the controllable separation of various organic liquid mixtures. The new-defined interfacial physical parameter (surface energy component index) performed well in all the wetting cases in this work. Furthermore, the facile method can be used in diverse applications including in-situ operation of extraction or back extraction, wettability patterning, organic reagent purifying, and the composition determination. To the best of my knowledge, it is the first time I have seen a membrane with tunable wettability for organic liquids. The idea of this work is interesting and original. The experiments are reasonable and the manuscript is well written. I believe the wetting theory presented would provide great help for researchers to design intelligent surfaces with adjustable wetting behaviors. Thus, I recommend the acceptance of the manuscript after addressing the following minor issues.

1. In the section of "Fabrication of the reconfigurable membranes", the multi-re-entrant hierarchical structure can be observed by the top SEM images of the FS-SSM. I think it would be better to further describe such a structure by an additional cross-section image, which should be captured and added to the Supplementary Information.
2. The authors have used complete techniques to characterize the membranes. For the description of the original substrate, besides the pore size as exhibited in the manuscript, I think it is necessary to measure the membrane thickness.
3. The as-prepared membrane shows the tunable wettability for various liquids. It is an exciting wetting phenomenon. How the wetting behaviors of the SSM substrate perform for the organic liquids? In order to eliminate the influence of the substrate, please provide the contact angles of typical polar protic liquid, polar aprotic liquid, and nonpolar liquid (such as water, DMF, and DOD) on the SSM substrate.
4. The authors have explained the corresponding fluxes of the membranes for different organic liquids in detail. To make it easy for readers to follow, please add the method description of the permeation flux test in the section of "Methods".

Detailed reply to reviewers' comments:

Reviewer #1:

Comment 1: In Fig. 1a, the high Ψ^p molecules are expressed in the form of a rod. In fact, most of the polar molecules are bent like water molecules rather than the rod shape. It would be easier to understand if you express it with a curved shape and add a color to express the polarity.

Author answer: Thanks very much for the referee's valuable suggestion.

In order to clearly demonstrate the structure of most high Ψ^p molecules, we have changed the corresponding schematic illustration from the original rod shape to the curved shape. Besides, an extra red color is added to express the corresponding polarity (Fig. 1).

For your convenience, the revised content in our manuscript is as follows:

P5/L97 in the revised manuscript:

Fig. 1 Schematic illustration of the reconfigurable membrane with tunable OL wettability. **a** The microstructure of the reconfigurable membrane. **b** On-demand OL separation procedures: the reconfigurable membrane demonstrates the lyophobicity for the low polar SECI (Ψ^p) liquids and the lyophilicity for the high polar SECI liquids.

After the joystick liquid (ethanol) triggers, the membrane will adopt an opposite wetting mode to accomplish OL separation based on the difference of dispersive SECI (ψ^d).

Comment 2: In Fig. 1, both the fluorosurfactant and the polyacrylate molecule of the adhesion layer are represented in green, which is confused. It would be good to modify the color so that they can be distinguished.

Author answer: Thanks very much for the referee's helpful suggestion.

To clearly distinguish the fluorosurfactant and the polyacrylate molecule of the adhesion layer, we have changed the color of the polyacrylate molecule of the adhesion layer from the original green to light blue in all corresponding schematic illustration (Fig. 1, Fig. 2a, Supplementary Fig. 1d, Supplementary Fig. 10, Supplementary Fig. 11a, and Supplementary Fig. 18a).

The revised Fig. 1 is as shown above, and other revised content is as follows:

P7/L148 in the revised manuscript:

Fig. 2 Fabrication process and characterizations of reconfigurable membranes. **a** Schematic diagram of the preparation procedures (G-SSM represents the polyacrylate glue coated SSM). **b** AFM images with height profile along the dotted line. **c** SEM images of the as-prepared membrane showing the hierarchical microstructures. **d** XRD pattern of FS-SSM and untreated SSM substrate compared with the respective standard patterns of TiO₂ anatase. **e** XPS wide spectra of FS-SSM and SSM, where “F” and “N”

refer to the Capstone FS-50, and “Ti” refers to TiO₂. **f** FTIR spectra of the membranes in different preparation stages.

P2 in the revised Supplementary Information:

Supplementary Figure 1. The chemical structure of the used reagents and the schematic illustration of the self-assembly and adhesive mechanism. **a**, Capstone FS-50 is one kind of commercial amphoteric fluorosurfactant that the main chemical structure is derived from betaine. The corresponding chemical structure formula is inferred from the product description and the characterization methods in this work. **b**, The main active chemical composition of the used adhesive is crosslinking polyacrylate that is grafted by special groups as described in the product description and the relevant literature.¹ **c**, The hydrogen bond self-assembly process is performed between the surface of the TiO₂ nanoparticle and the fluorosurfactant molecule. **d**, The coating works in a synergistic way based on the two components. Mechanical interlocking occurs as the crosslinking adhesive nanoparticles flow into the pores on the SSM and the gaps between the composite TiO₂ nanoparticles. On the one hand, the interlocking improves the strength and durability of the as-prepared coating. On the other hand, it also serves to increase the total contact surface area between different nanoparticles to form the multi-re-entrant hierarchical structure, which is beneficial for enhancing the surface wettability of the coated membrane in Cassie-Baxter theory.

P11 in the revised Supplementary Information:

Supplementary Figure 10. Hierarchical tiers of the fabricated surfaces and the corresponding wettability. Based on the SEM images (scale bar is 50 μm) and corresponding schematics, the mesh substrate possesses one tier of surface hierarchy compared with the smooth substrate. The amphoteric fluorosurfactant is dip-coated on the mesh to realize two tiers of surface hierarchies. Three tiers of surface hierarchies are further prepared by the spray-coating method with the paint only. Whereas with the “glue + paint” co-spray-coating method, the FS-SSM features the four tiers of surface hierarchies which significantly reduces the liquid-solid contact area. As exhibited in the contact angle profiles, the increased tiers of hierarchies can promote the repellence to the probe liquid (cyclohexane), indicating the crucial role of the multi-re-entrant hierarchical structure.

P12 in the revised Supplementary Information:

Supplementary Figure 11. The rapid response of different wetting behaviors of the FS-SSM. **a**, Images illustrate a jet of toluene bouncing off the membrane surface without residual liquid based on the proposed explanation of low-adhesion quasi-superlyophobic surface. **b**, Time-sequence photographs of toluene droplet bouncing and water wetting on the membrane (the contact moment of the liquid droplet and the solid surface is defined as 0). The nonpolar toluene droplet bounced up in a short time without wetting the surface, indicating the weak interaction between toluene and FS-SSM. On the contrary, the water droplet spread rapidly on the surface due to the strong affinity. Droplet sizes, $\sim 6 \mu\text{L}$.

P19 in the revised Supplementary Information:

Supplementary Figure 18. The characterization of surface morphology and chemical composition of the PR-SSM. **a**, Schematic illustration of the achievement of the opposite wetting behaviors through particle rearrangement. **b**, SEM images of the PR-SSM. The microstructure identifies the reassembly result of TiO₂ nanoparticles intuitively, as evidenced by the produced larger scale of spherical nanoparticle aggregation after the treatment of ethanol. **c**, Variation of element content between FS-SSM and PR-SSM. Compared with the original FS-SSM, increased carbon element and a sharp decrease of the fluorine element are detected to be distributed on the surface of PR-SSM, highlighting the exposure of glue molecules on the nanoparticles' interfaces. **d**, XPS wide spectra of PR-SSM. The increased C signal and decreased F signal of the membrane after the ethanol trigger convincingly proves the process of particle rearrangement. **e**, FTIR spectra of PR-SSM. The characteristic peaks of the PR-SSM coincide with the corresponding peaks of G-SSM, meaning that polyacrylate structure can be strongly detected on account of the migration process of glue molecules.

Comment 3: In of Supplementary Figure 18, it is recommended to perform the long-term reusability experiment more than 30 cycles because it usually shows a phenomenon of poor durability from 10 cycles or more.

Author answer: Thanks very much for the referee's professional suggestion.

Aiming to reliably evaluate the long-term reusability of both FS-SSM and PR-SSM, more separation cycles for formamide/dodecane mixture (representative sample) were performed which were increased from 10 to 35. The corresponding fluxes and separation efficiencies were added to **Supplementary Fig. 22**. It can be observed that the FS-SSM exhibited stable permeation flux ($> 3,300 \text{ L m}^{-2} \text{ h}^{-1}$) and high separation efficiency ($> 99.0\%$) even after 35 cycles. As for the PR-SSM, the separation efficiency is as high as 98.5% after the same 35 cycles. Although the permeation flux has a

decrease from the initial state, it still maintains a relatively high value of $3,400 \text{ L m}^{-2} \text{ h}^{-1}$. All the results prove the excellent stability and recyclability of the as-prepared membranes.

The revised content in our manuscript is as follows:

P13/L304-306 in the revised manuscript:

Such a separation process can be repeated at least **35 cycles** without apparent separation efficiency decline and permeation flux loss, indicating the excellent long-term reusability (Supplementary Fig. 22a).

P14/L322-324 in the revised manuscript:

The membrane exhibits high liquid fluxes (above $4,000 \text{ L m}^{-2} \text{ h}^{-1}$) and outstanding cyclic performance (**over 35 cycles**) during the **continuous separation process** (Figs. 4e, f, and Supplementary Fig. 22b).

P23 in the revised Supplementary Information:

Supplementary Figure 22. The cycle stability experiments of the FS-SSM and PR-SSM. **a**, To assess the long-term reusability of the FS-SSM, cycle stability experiments were performed for **35 cycles**, taking formamide/dodecane mixture as the representative sample. On the one hand, the as-prepared mesh exhibits a stable permeation flux with no significant decline during **35 cycles**, demonstrating that the as-prepared membrane is equipped with an antifouling property. On the other hand, high separation efficiency and low insoluble phase residual remained in separation cycles, exhibiting excellent durability in long-term work. **b**, Change of the separation efficiency and permeation flux of the PR-SSM as separating formamide/dodecane mixture **for 35 cycles**. **The membrane shows superior separation efficiency ($> 98.5\%$) during the continuous separation process. Moreover, the permeation flux is still above $3,400 \text{ L m}^{-2} \text{ h}^{-1}$ after 35 cycles, indicating superior cycle stability of the membranes.**

Comment 4: Separation efficiency is generally lowest in first cycle and high in other cycles, but the separation efficiency in Supplementary Figure 18 shows a different trend. Please explain why.

Author answer: Thanks very much for the referee’s precious question. We are sorry for your confusion due to the previous unclear description.

In this work, the separation efficiencies of OL mixtures were measured by a general method based on the mass ratio of the permeable liquid before and after the separation. Hence, the separation efficiency of the initial membrane separation would be significantly lower than other cycles due to the absorption of the permeable liquid. In fact, to eliminate the influence mentioned above, we start recording the separation efficiency after the initial separation cycle during the cycle stability experiments. The membranes can reach a saturated adsorption state after such an operation. The similar method was also used in previous reports (For example, J. Li, et al. *J. Mater. Chem. A* **2015**, 3, 14696; J. Li, et al. *Chem. Eng. J.* **2016**, 287, 474-481; S. Oh, et al. *Langmuir* **2019**, 35, 7769-7782, and so on), which were added in this revised version.

To illustrate this method more intuitively, the following Figure shows the separation performance of the two as-prepared membranes in previous separation cycles that also contains the initial cycle (set as 0 cycle). The variation trend of the separation efficiencies is as same as the aforementioned phenomenon and the proposed comment. Except that the separation efficiency of the initial cycle is as low as about 94.0%, the other cycles exhibited stable and outstanding separation efficiencies for either FS-SSM or PR-SSM. Therefore, the initial liquid wetting or separation can effectively eliminate the influence of the absorption for efficiency fluctuations. The subsequent measurements will more objectively evaluate the actual reusability performance of the membranes. The corresponding description of the separation efficiency test is further explained in the “Methods” section of the revised manuscript.

Fig. 1 of Response The separation efficiencies of **a**, the FS-SSM and **b**, the PR-SSM during different separation cycles, taking formamide/dodecane mixture as the representative sample.

The revised content in our manuscript is as follows:

P20/L471-473 in the revised manuscript:

The OL separation efficiency was measured three times in one separation cycle, calculated in terms of the liquid rejection coefficient (η_m) from the following equation:

$$\eta_m = \left(1 - \frac{m_t}{m_o}\right) \times 100\% \quad (6)$$

Where m_o and m_t represent the mass of permeable liquid before and after the separation. To eliminate the influence of the membrane absorption, the separation efficiencies were calculated after the initial separation.⁴⁶⁻⁴⁸

P27 in the revised manuscript:

46. Li, J. et al. Underwater superoleophobic palygorskite coated meshes for efficient oil/water separation. *J. Mater. Chem. A* **3**, 14696-14702 (2015).
47. Li, J.-J., Zhu, L.-T., & Luo, Z.-H. Electrospun fibrous membrane with enhanced switchable oil/water wettability for oily water separation. *Chem. Eng. J.* **287**, 474-481 (2016).
48. Oh, S. et al. Performance analysis of gravity-driven oil-water separation using membranes with special wettability. *Langmuir* **35**, 7769-7782 (2019).

Comment 5: In order to reinforce the originality of the paper, additional experiments are required to prove the mechanism or principle for the change of the membrane properties by joystick liquid. This is because papers that change the properties of membranes using joystick liquids have already been reported.

Author answer: Thanks very much for the referee's insightful suggestion.

As reminded by the reviewer, it has been reported that the surface properties of the membranes would be changed by joystick liquids treatment. The corresponding works had been cited in our previous version of the manuscript to assist in explaining the mechanism (**Ref. 39:** L. Li, et al. *Adv. Mater.* **2017**, *29*, 1702517; **Ref. 40:** Z. Li, et al. *Matter* **2019**, *1*, 1-13). However, previous works focused on the enhancement of water repellency of the as-prepared materials, which can improve the surface roughness of the uniform coating using joystick liquids. Here our membrane possessed two synergistic and heterogeneous layers with opposite surface energy components, which can exhibit positive/negative wetting regularity for various OLs by triggering different layer configurations using joystick liquids. Although both use joystick liquids to change the surface properties, there are great differences in microstructure construction, surface energy control, and applications.

The mechanism of the wetting transition phenomenon of our membranes can be attributed to the migration of the polyacrylate layer below the paint nanoparticles via the joystick liquid wetting. In our previous version of the manuscript, such a principle was proved by the characterization of surface morphology and chemical composition between the initial state (FS-SSM) and the final state (PR-SSM), including SEM images, element content comparison, XPS spectra, and FTIR spectra. Compared with the FS-SSM, the corresponding signal of the polyacrylate layer can be strongly detected on the PR-SSM surface after the joystick liquid treatment in the aforementioned characterization.

We greatly agree with the referee's suggestion about the additional proof experiments. In addition to the comparison between the initial and final states, it is necessary to further investigate the dynamic migration process of the polyacrylate layer. Here a laser confocal microscope was used to characterize such a migration process.

The laser confocal microscope uses fluorescent species to acquire the sample images at varying focal depths. We first analyzed the fluorescence emission spectra of the paint nanoparticles, the used polyacrylate adhesive, and the as-prepared FS-SSM. As shown in **Supplementary Fig. 19**, it can be seen that intense fluorescence peaks (the blue band, 420 ~ 450 nm) were strongly observed on the adhesive coated SSM (the substrate coated only by the polyacrylate adhesive) ($\lambda_{\text{ex}} = 405 \text{ nm}$). In contrast, the membrane that was only sprayed by the paint nanoparticles exhibited no obvious fluorescence peaks under the same excitation condition. As for the FS-SSM, the corresponding fluorescence peaks of the adhesive were significantly weakened by the paint nanoparticles loaded on the upper layer. Hence, the aforementioned results prove the feasibility of using a laser confocal microscope, which supports us in utilizing the change of the fluorescence intensity of the polyacrylate layer to study the migration process.

The confocal images of the FS-SSM at different stages via joystick liquid treatment were captured as exhibited in **Supplementary Fig. 20**. In the initial stage, weak fluorescence of the polyacrylate layer was clearly observed on the FS-SSM surface (**Supplementary Fig. 20a**). As the treatment time of the joystick liquid increases, the membrane exhibited obvious fluorescence enhancement, indicative of the gradual migration process of the polyacrylate layer (**Supplementary Fig. 20b-g**). These visible results are in good agreement with the quantitative fluorescence intensity data in **Supplementary Fig. 20h**. The quantitative data demonstrated an increase in fluorescent strength ($p < 0.05$, in contrast to the initial state), corresponding to the migrating polyacrylate amount on the surface. Hence, we successfully proved our proposed mechanism through the collaborative characterization of the static comparison and dynamic process.

Thank you again for your recognition and important suggestion.

The revised content in our manuscript is as follows:

P12/L256-265 in the revised manuscript:

Such a mechanism is supported by the systematical characterization of surface morphology and chemical composition **that the corresponding signal of the polyacrylate layer can be strongly detected on the PR-SSM surface after the joystick liquid treatment (Supplementary Fig. 18). The dynamic migrating process is further demonstrated by utilizing a laser confocal microscope. The changing visible blue fluorescence of the polyacrylate layer can be clearly observed in the confocal images of the as-prepared membrane via joystick liquid treatment for different time (Supplementary Figs. 19 and 20a-g). It can be seen that the fluorescence intensity gradually increases with treatment time, suggesting the dynamic migrating process of the polyacrylate layer triggered by the joystick liquid (Supplementary Fig. 20h).**

P22/L516-518 in the revised manuscript:

The fluorescence spectra were measured by a fluorescent photometer (RF-2000, SHIMADZU, Japan). Confocal images were collected by a laser confocal microscopic system (LSM-780, Zeiss, Germany).

P20 in the revised Supplementary Information:

Supplementary Figure 19. Fluorescence spectra of different membranes including the adhesive coated SSM, paint coated SSM, and the final FS-SSM ($\lambda_{\text{ex}} = 405 \text{ nm}$). The used polyacrylate adhesive exhibits strong blue fluorescence (420 ~ 450 nm), while no fluorescence is recorded with the paint nanoparticles.

P21 in the revised Supplementary Information:

Supplementary Figure 20. Demonstration of the dynamic transition process of the FS-SSM surface via joystick liquid triggering. **a-g**, The confocal images of the FS-SSM after different treatment time of the joystick liquid (30 s interval). “0 s” and “180 s” represents the stages of FS-SSM and PR-SSM, respectively. Each sample was tested on three different positions and taken the average ($\lambda_{\text{ex}} = 405 \text{ nm}$). **h**, Fluorescent intensities of the FS-SSM after different treatment time of the joystick liquid, $p < 0.05$ between the initial membrane and other states, data are represented as mean \pm standard deviation (SD).

Reviewer #2:

In this manuscript, the authors developed a nanoparticle-embedded gating membrane composed of different reconfigurable layers, which can realize the controllable separation of various organic liquid mixtures. The new-defined interfacial physical parameter (surface energy component index) performed well in all the wetting cases in this work. Furthermore, the facile method can be used in diverse applications including in-situ operation of extraction or back extraction, wettability patterning, organic reagent purifying, and the composition determination. To the best of my knowledge, it is the first time I have seen a membrane with tunable wettability for organic liquids. The idea of this work is interesting and original. The experiments are reasonable and the manuscript is well written. I believe the wetting theory presented would provide great help for researchers to design intelligent surfaces with adjustable wetting behaviors. Thus, I recommend the acceptance of the manuscript after addressing the following minor issues.

Comment 1: In the section of “Fabrication of the reconfigurable membranes”, the multi-re-entrant hierarchical structure can be observed by the top SEM images of the FS-SSM. I think it would be better to further describe such a structure by an additional cross-section image, which should be captured and added to the Supplementary Information.

Author answer: Thanks very much for the referee’s affirmation of our work and important suggestion.

In order to characterize the morphology more clearly, the cross-sectional SEM images of the FS-SSM membrane have been added as **Supplementary Fig. 6**. The membrane exhibited visually the multi-re-entrant hierarchical structure composed of the paint layer and the adhesion layer.

For your convenience, the revised content in our manuscript is as follows:

P6/L127-132 in the revised manuscript:

The scanning electron microscopy (SEM) images demonstrate the hierarchical textures composed of stacking micro-/nanoscale structures with multi-re-entrant curvatures attributed to the cooperative aggregates of adhesive nanoparticles and composite TiO₂ nanoparticles in mechanical interlocking mode (Fig. 2c, Supplementary Fig. 1d). Besides, the cross section of the FS-SSM is also observed as shown in **Supplementary Fig. 6**, indicating the aforementioned bilayer composite structures.

P7 in the revised Supplementary Information:

Supplementary Figure 6. The cross-sectional SEM images of the FS-SSM membrane. The two superimposed layers can be clearly observed that contains the paint layer and the adhesion layer.

Comment 2: The authors have used complete techniques to characterize the membranes. For the description of the original substrate, besides the pore size as exhibited in the manuscript, I think it is necessary to measure the membrane thickness.

Author answer: Thanks very much for the referee's valuable suggestion.

In order to characterize the thickness of the SSM substrate, the vernier caliper measurement and the cross-sectional SEM images of the SSM substrate have been added and observed in **Supplementary Fig. 4c**. According to the corresponding results, the thickness of the substrate is about 80 μm.

The revised content in our manuscript is as follows:

P6/L120-123 in the revised manuscript:

For further preparing the targeted membrane, the commercial polyacrylate adhesive (Supplementary Fig. 1b) was sprayed homogeneously onto the stainless steel mesh (SSM, the average pore size is ~ 50 μm, the thickness is ~ 80 μm, Supplementary Fig. 4) to form a glue layer bonding tightly the functional paint.

P5 in the revised Supplementary Information:

Supplementary Figure 4. a, Top image and b, high-magnification image of the SSM substrate. The wire spacing is ~ 50 μm, and the diameter of the individual wire is ~ 30

μm . **c**, The cross-sectional SEM image and the corresponding vernier caliper measurement of the SSM substrate, indicating that the membrane thickness is $\sim 80 \mu\text{m}$.

Comment 3: The as-prepared membrane shows the tunable wettability for various liquids. It is an exciting wetting phenomenon. How the wetting behaviors of the SSM substrate perform for the organic liquids? In order to eliminate the influence of the substrate, please provide the contact angles of typical polar protic liquid, polar aprotic liquid, and nonpolar liquid (such as water, DMF, and DOD) on the SSM substrate.

Author answer: Thanks very much for the referee's valuable suggestion.

The contact angles of the typical OLs (water, DMF, and DOD) on the substrate were measured as exhibited in **Supplementary Fig. 9**. The substrate shows lyophobicity for water and superlyophilicity for DMF and DOD, consistent with classical interfacial theory. The reason is that most OLs (such as DMF and DOD) have low surface energies, while water has high surface energy (72.8 mJ m^{-2}). The inherent wetting behaviors of the substrate are completely different from the FS-SSM, indicating the tunable wettability is mainly determined by the reconfigurable layers rather than the SSM substrate.

The revised content in our manuscript is as follows:

P8/L168-169 in the revised manuscript:

Besides, it can be seen that the membrane is particularly distinctive in the interaction with polar aprotic liquids. For example, N,N-dimethylformamide embodies superlyophilicity on the membrane surface while the opposite wettability for nitromethane, though both have incredibly similar SE values (36.5 mJ m^{-2} and 36.8 mJ m^{-2} , respectively). The influence of the substrate on the aforementioned unique wettability can be ignored as shown in **Supplementary Fig. 9**.

P10 in the revised Supplementary Information:

Supplementary Figure 9. The contact angles of water, dimethylformamide, and dodecane on the SSM substrate. The substrate exhibits different wettability for the typical OLs compared with FS-SSM, suggesting the coated composite layers play a

crucial role in the final unique wetting behaviors for various liquids rather than the SSM substrate.

Comment 4: The authors have explained the corresponding fluxes of the membranes for different organic liquids in detail. To make it easy for readers to follow, please add the method description of the permeation flux test in the section of “Methods”.

Author answer: Thanks very much for the referee’s valuable suggestion.

The method of the permeation flux test of the membranes has been added to the Methods in detail.

The revised content in our manuscript is as follows:

P21/L473-478 in the revised manuscript:

The permeation flux of the membrane was determined by calculating the volume of the separated phase in unit time using the following equation:

$$F = \frac{V}{At} \quad (7)$$

Where F is the flux of the membrane, V is the volume of the separated phase, A is the effective area of the membrane (3.80 cm^2), and t is the testing time. Each data point was the average value of three parallel experiments.

REVIEWERS' COMMENTS

Reviewer #1 (Remarks to the Author):

The originality of this paper can be represented in three constituents: TiO₂/FS-50, adhesive layer (polyacrylate), and Joystick liquid. I would like to inform you that I have focused on these three constituents and conducted an in-depth review on whether this paper is suitable for Nature communications.

First of all, the paper (Ref. 18: Adv. Funct. Mater. 28 (2018) 1706867), which capstone FS-50 was coated on TiO₂ and used as a separation membrane with an adhesive layer, is very similar to this paper. And another similar paper (NATURE COMMUNICATIONS 11 (2020) 425) showed a separation effect with only TiO₂ and FS-50 even without an adhesive layer. The separation membrane structure, composition and its preparing method of these two papers are too analogous to this paper. Moreover, polyacrylate used as an adhesive layer in this paper is not the first material the author used with TiO₂, it has been used as an adhesive layer with TiO₂ particles (Progress in Organic Coatings 85 (2015) 101–108, Polymer composites 39 (2018) 4467–4476 and Progress in Organic Coatings 112 (2017) 153–161). Therefore, among the three constituents of originality of this paper, joystick liquid is the most important constituent in determining whether this paper is acceptable to Nature communication. Unfortunately, this is not the first paper to perform separation using joystick liquid. Two papers (ACS Appl. Mater. Interfaces 10 (2018) 40265 and NATURE COMMUNICATIONS 11 (2020) 425) have already reported the separation membrane using joystick liquid.

Hence, I commented to the author that the originality of this paper should be enhanced, and the author supplemented this paper but was insufficient to improve the originality. Consequently, it is considered that it still does not have the originality suitable for Nature communications.

Reviewer #2 (Remarks to the Author):

After reading the revised manuscript and the response letter, I think that it can be accepted by NC as is.

Response to Referees

Reviewer #1:

Comment: The originality of this paper can be represented in three constituents: TiO₂/FS-50, adhesive layer (polyacrylate), and Joystick liquid. I would like to inform you that I have focused on these three constituents and conducted an in-depth review on whether this paper is suitable for Nature communications.

First of all, the paper (Ref. 18: Adv. Funct. Mater. 28 (2018) 1706867), which capstone FS-50 was coated on TiO₂ and used as a separation membrane with an adhesive layer, is very similar to this paper. And another similar paper (NATURE COMMUNICATIONS 11 (2020) 425) showed a separation effect with only TiO₂ and FS-50 even without an adhesive layer. The separation membrane structure, composition and its preparing method of these two papers are too analogous to this paper. Moreover, polyacrylate used as an adhesive layer in this paper is not the first material the author used with TiO₂, it has been used as an adhesive layer with TiO₂ particles (Progress in Organic Coatings 85 (2015) 101–108, Polymer composites 39 (2018) 4467-4476 and Progress in Organic Coatings 112 (2017) 153–161). Therefore, among the three constituents of originality of this paper, joystick liquid is the most important constituent in determining whether this paper is acceptable to Nature communication.

Unfortunately, this is not the first paper to perform separation using joystick liquid. Two papers (ACS Appl. Mater. Interfaces 10 (2018) 40265 and NATURE COMMUNICATIONS 11 (2020) 425) have already reported the separation membrane using joystick liquid.

Hence, I commented to the author that the originality of this paper should be enhanced, and the author supplemented this paper but was insufficient to improve the originality. Consequently, it is considered that it still does not have the originality suitable for Nature communications.

Author answer: Thanks very much for the referee's comments. Here we reply to the corresponding comments point by point as follows. In this revised manuscript, we also supplemented some statements to enhance the originality of our work.

Response to the originality summary:

The referee pointed out that the originality of this paper can be represented in three constituents containing the composite paint nanoparticles, the adhesive layer, and the joystick liquid trigger. However, we consider these three constituents were only summarized from the microstructure and preparation method, which is not the essential novelty of our work. In fact, the novelty of this work can be focused on and summarized from four points including the proposed interfacial theory, the distinct wettability performance, the controllable liquid-liquid separation, and the expansibility in applications.

Response to the nanoparticles evaluation:

As for the evaluation of the first constituent (the composite paint nanoparticles) in the comments, the paper (Ref. 18: Adv. Funct. Mater. 28 (2018) 1706867) indeed inspired us to construct the surface with different surface energy components. However, our work is entirely different from the aforementioned paper in terms of research direction, surface properties of the membrane, and the applications:

1. **From the perspective of the research direction**, in the paper (Ref. 18), the authors used the composite nanoparticles to obtain the superoleophobic/superhydrophilic surfaces, which focused on the oil/water wetting direction (the surface energy difference is as large as 60 mJ m^{-2}). Notably, our work utilizing two synergetic aggregation-induced layers with different SE components extends the scope of the application from the oil/water field to the more complex organic liquids field (a record-breaking surface energy difference as small as 0.3 mJ m^{-2}).
2. **From the perspective of the surface wetting property**, Ref. 18 focused on the specific and constant wettability of oil or water in a different wetting mode from the previous reports. In comparison, our work aims to achieve the wetting switch and manipulative liquid-repellent properties for various organic liquids, which is a more intelligent surface property. No publications can realize such a tunable wetting phenomenon as far as we know.
3. **From the perspective of the applications**, the two works are also quite different. Ref. 18 realized the single-mode oil transport, oil-water separation, and emulsion demulsification (oil-water system). Our work, in addition to the controllable and universal liquid-liquid separation (at least 22 types of oily liquids system), can be extended to other novel applications including in-situ extraction or back extraction, micro-droplet manipulation, chemical reaction purification, and composition detection sensors that cannot be realized by common superwetting materials.

For the other paper (NATURE COMMUNICATIONS 11 (2020) 425), we also conducted in-depth reading and discovered that this work is completely different from our work from the following points. In order to make readers understand this field more clearly, this excellent paper has been cited as Ref. 24 in the revised manuscript.

1. **From the perspective of the used materials, membrane structure, and the preparation method**, the two works are not as similar as the referee stated, or even that there is no mutual connection. The above-mentioned paper presented a self-assembly strategy on the construction of supramolecular frameworks composing of inorganic polyanionic clusters, cationic organic hosts, and a bridging guest. It is in contradiction with the referee's statement that the works use the same $\text{TiO}_2/\text{FS-50}$.
2. **From the perspective of the wetting principle**, the gel membrane of the above-mentioned paper exhibited hydrophobicity and oleophilicity, which was a constant surface wetting property. In contrast, our membrane has switchable wettability for various liquids. More importantly, we provide a general design principle of any surface with the target or tunable wettability for extensive organic liquids.
3. **From the perspective of the separation mechanism**, the two types of membranes work differently. The gel membrane utilized the liquid-infused method assisted with ethanol to achieve the switchable separation mode, in which the infused immiscible

liquid layer acted as the repellent surface rather than the covalently modified solid surface with tunable properties. While our membrane realizes the controllable separation based on the positive/negative wetting regularity by triggering different layer configurations using joystick liquids, which does not need the prewetting operation.

Response to the adhesive layer evaluation:

As stated by the referee, it is not the first time that the polyacrylate was used with TiO₂ nanoparticles. However, it is the first time to use such materials in the field of surface wettability control for organic liquid. As for the three listed references, we further discuss and emphasize the main differences from our work as follows.

1. **The research directions are different.** All three references prepared composite latex applied in leather finishing or thermal insulation coating, aiming to improve the thermal stability, anti-yellowing, mechanical or antibacterial properties of the coatings. Nevertheless, our work focuses on the wettability design and the universal interfacial theory, which can be applied in the liquid-liquid separation.
2. **The reasons for the selection of the materials are different.** The listed works used the inorganic particles (TiO₂ nanoparticles) and polymers (polyacrylate) to enhance the overall performance in mechanical, electrical, photonics properties, and so on. But in our work, we are concerned about the amphoteric fluorosurfactant rather than the TiO₂ nanoparticles due to the distinct surface energy components. In fact, the used TiO₂ nanoparticles play the role of loading the fluorosurfactant. The tunable surface energy components can be further achieved by introducing the adhesive layer (polyacrylate) with opposite configurations.
3. **The preparation methods are different.** The listed references prepared the homogeneous TiO₂ nanoparticles/polyacrylate coating by the in-situ polymerization. In contrast, our membrane possesses two synergistic and heterogeneous layers (nanoparticle layer and adhesive layer) by mechanical interlocking, which can easily regulate the surface properties based on such a reconfigurable microstructure.

Response to the joystick liquid evaluation:

It is true that the joystick liquid trigger is an important part of this work because it is the key to induce surface reconstruction with tunable surface energy components. Although the two listed references realized the liquid separation by the assistance of ethanol, our membrane works with a completely different mechanism for a broader range of liquid systems (not just oil/water systems). For the listed references, the joystick liquid worked as the infused layer to make the blocked phase pass through and would not change the surface properties. In contrast, in our work, the joystick liquid acts as an external stimulus to change the chemical property and physical structure of the surface. Such a surface regulation, especially intrinsic wettability, is of great significance in extensive research fields.

In the paper (ACS Appl. Mater. Interfaces 10 (2018) 40265), the membrane was essentially superhydrophobic and only suitable for oil/water systems. In order to achieve the switchable oil/water separation, the membrane was supposed to be prewetted by ethanol in which a certain amount of ethanol molecules can be adsorbed

into the micropores. In this case, the water can be miscible with an ethanol layer and the membrane would exhibit the superhydrophilicity. However, the operation would induce the existence of ethanol in the filtrate during the separation that is more difficult to separate from the mixtures. Compared with the aforementioned paper, our work avoids the additional prewetting process and the miscible case due to the intrinsic surface properties of our membrane.

For the paper (NATURE COMMUNICATIONS 11 (2020) 425), we have explained the corresponding difference compared with our work in the part of “Response to the nanoparticles evaluation”. Moreover, the liquid-infused problem can also be solved by our membrane with changeable surface energy components.

In summary, by combining the three proposed constituents, we obtained a smart superwetting membrane for various liquids with different surface energy which is not achieved before. Last but not least, we develop a new-defined physical parameter that can classify full-SE-range liquids using a more reasonable way, and liquid wetting or anti-wetting on the resulting surfaces exhibits a surprisingly straightforward connection with this physical parameter. We think it is a new contribution to the specific wettability design for broad applications. Hence, we believe the originality of this work can be supported by the new-developed interfacial theory, the distinct wettability performance, the controllable liquid-liquid separation, and the expansibility in applications.

Reviewer #2:

Comment: After reading the revised manuscript and the response letter, I think that it can be accepted by NC as is.

Author answer: Thanks very much for your affirmation of our work. We also appreciate your kind recommendation for our paper to be published without further revisions.